# Maximizing the Clinical Value of Blood-Based Biomarkers for Mild Traumatic Brain Injury

**DOI:** 10.3390/diagnostics13213330

**Published:** 2023-10-28

**Authors:** Steven H. Rauchman, Aaron Pinkhasov, Shelly Gulkarov, Dimitris G. Placantonakis, Joshua De Leon, Allison B. Reiss

**Affiliations:** 1The Fresno Institute of Neuroscience, Fresno, CA 93730, USA; dr.rauchman@yahoo.com; 2Department of Medicine and Biomedical Research Institute, NYU Grossman Long Island School of Medicine, Mineola, NY 11501, USA; aron.pinkhasov@nyulangone.org (A.P.); shellygulk1234@gmail.com (S.G.); joshua.deleon@nyulangone.org (J.D.L.); 3Department of Neurosurgery, NYU Grossman School of Medicine, New York, NY 10016, USA; dimitris.placantonakis@nyulangone.org

**Keywords:** concussion, mild traumatic brain injury, biomarker, diagnosis, GFAP, rehabilitation

## Abstract

Mild traumatic brain injury (TBI) and concussion can have serious consequences that develop over time with unpredictable levels of recovery. Millions of concussions occur yearly, and a substantial number result in lingering symptoms, loss of productivity, and lower quality of life. The diagnosis may not be made for multiple reasons, including due to patient hesitancy to undergo neuroimaging and inability of imaging to detect minimal damage. Biomarkers could fill this gap, but the time needed to send blood to a laboratory for analysis made this impractical until point-of-care measurement became available. A handheld blood test is now on the market for diagnosis of concussion based on the specific blood biomarkers glial fibrillary acidic protein (GFAP) and ubiquitin carboxyl terminal hydrolase L1 (UCH-L1). This paper discusses rapid blood biomarker assessment for mild TBI and its implications in improving prediction of TBI course, avoiding repeated head trauma, and its potential role in assessing new therapeutic options. Although we focus on the Abbott i-STAT TBI plasma test because it is the first to be FDA-cleared, our discussion applies to any comparable test systems that may become available in the future. The difficulties in changing emergency department protocols to include new technology are addressed.

## 1. Introduction

Traumatic brain injury (TBI) is a neurophysiologic event experienced by an estimated 1.5 million Americans every year according to the Centers for Disease Control and Prevention (CDC) [1]. Mortality across all TBI severities is approximately 3%. While TBI is well known to cause cognitive, motor, and behavioral symptoms, both acutely and as long-term complications, its morbidity is difficult to quantify. Although most TBI is characterized as mild, even these less severe head injuries can bring long-term sequelae [2]. Understanding the severity of TBI and establishing proper diagnostic methods for TBI is critical for the patient’s functional recovery and for structuring a personalized rehabilitation plan [1,2]. Traumatic axonal injury (TAI) comes with high variability in type and degree of neurological impairment, with injury location a significant determinant of prognosis [3]. TAI refers specifically to damage to axons in the white matter that causes microscopic pathologic brain lesions and manifests as axonal tearing, axonal swelling, and impairment of transport along the axon [4,5]. This type of damage comes about due to the rotational and acceleration/deceleration movements of the head during TBI [6]. TAI has been documented at autopsy in persons who have sustained mild TBI and died of other causes [7]. Newer imaging techniques allow diagnosis of TAI in mild TBI, and the use of blood biomarkers to detect the presence of TAI in persons with mild TBI will be addressed later in this review.

There has been growing concern about the missed identification of mild TBI and the risk of developing intracranial lesions that could require neurosurgical intervention [8]. Protocols for managing mild TBI are a work in progress, with an increasing focus on brain injury biomarkers for prognostic utility [9]. Although some of these biomarkers correlate with mortality and outcome, further research on their clinical utility needs to be conducted [10,11].

This review focuses on the current methods of TBI assessment and the need for better predictive models that are easy to employ in the clinical setting. We propose that, via data aggregation, measures of complementary blood biomarkers can provide value beyond their recommended post-injury window, which may be as narrow as 12 h [12,13]. The concept behind this would be the use of voluminous data collected during clinical TBI assessment encompassing a physical exam, imaging, and blood biomarkers along with longitudinal follow-up and outcome to produce robust and replicable results that can provide insight into heretofore-unrecognized factors in TBI recovery and perhaps extend the usefulness of the test beyond the current sensitivity limit despite the rapid fall in biomarker levels over time.

## 2. Current Modes of Mild TBI Evaluation

### 2.1. Glasgow Coma Scale

Early and accurate TBI diagnosis is important for the detection of any brain injuries that may lead to long-term issues. However, the heterogeneity of TBI signs and symptoms makes characterization difficult. A tool that is often used clinically is the Glasgow Coma Scale (GCS), which sums scores from three categories including best eye response, best verbal response, and best motor response to yield a score between 3 and 15, with scores between 13 and 15 considered a mild TBI [14]. Although the GCS is a solid basic classifier of TBI severity, it is not perfect and may lead to underestimation of TBI damage [15].

### 2.2. Imaging by Computed Tomography

Various imaging methods such as computed tomography (CT) and magnetic resonance imaging (MRI) can be used promptly to determine structural damage [16]. Non-contrast CT is one modality for diagnosing TBI that is available in acute settings [17]. In the emergency department, clinicians can decide whether to perform a non-contrast brain CT after completing various tests of neurocognitive function, visuomotor function, and balance [18]. CT is often normal in mild TBI and is largely used to visualize the presence and location of both extra-axial and intra-axial hemorrhages [19,20,21]. The development of novel computer-aided diagnosis systems that utilize non-contrast axial CT brain images to detect hematoma, raised intracranial pressure, and midline shift can be useful for the early management of TBI [22,23]. Some shortcomings of CT scans are the underestimation of parenchymal contusions and limitations in detecting diffuse axonal injury [24,25]. Diffuse axonal injury is an important aspect of TBI pathology. It is the process in which there is axonal damage due to acute or repetitive TBI leading to deficits in cerebral connectivity that may or may not recover [26]. Axonal injuries result from axonal stress and predominately affect white matter tracts and are recognized to occur with milder TBI injuries to a lesser extent [27]. Effectively diagnosing this injury is essential because axonal injury is an important determinant of long-term effects of neurotrauma, and deeper lesions, particularly in the brainstem and corpus callosum, as well as higher-grade lesions, are associated with poorer cognitive and functional outcomes [28,29,30].

Widely used CT-based classification systems for standardized interpretation of brain images include the Helsinki, Rotterdam, and Stockholm systems. Rotterdam scores are determined by the status of the basal cisterns, whether there is a midline shift, the presence of an epidural mass lesion, and the detection of intraventricular or subarachnoid hemorrhage [31,32]. The Stockholm score uses midline shift as a continuous variable, a separate scoring for traumatic subarachnoid hemorrhage, and is the only scoring system that takes diffuse axonal injury into consideration [33]. The Helsinki scoring system looks at lesion size and type, whether there is intraventricular hemorrhage and the condition of suprasellar cisterns [34]. A study by Biuki et al. on 171 blunt TBI patients aged above 15 years admitted to a hospital in Iran analyzed the specificity, sensitivity, and negative/positive predictive value for Helsinki, Rotterdam, and Stockholm scores. The study concluded that the Rotterdam was superior in predicting death in TBI patients, while the Helsinki score was more sensitive and had a higher negative predictive value for predicting the six-month outcome. The authors propose that the Rotterdam score would be best used for predicting death and the Helsinki score to predict the six-month outcome [35].

### 2.3. Magnetic Resonance Imaging

MRI is a noninvasive imaging technique that uses non-ionizing electromagnetic radiation to provide useful information on brain anatomy and structure, while functional MRI can go further by capturing changes in brain function, blood flow, and connectivity [36,37,38]. MRI has a higher sensitivity than CT and is useful in mild TBI to detect subtle findings such as small lesions, contusions, and intracranial bleeds [39].

Despite the greater accuracy of MRI, there is a growing concern about how useful neuroimaging is as a prognostic marker for TBI [40]. A study on the prognostic impact of MRI-defined diffuse axonal injury after TBI that had a sample of 311 patients and assessed quality of life and three-year mortality found that neither clinical nor anatomical diffuse axonal injury was related to survival, Glasgow Outcome Scale-Extended, or quality of life scores [41]. The study’s limitations are that it failed to include brain injury patterns such as subdural/epidural hemorrhage, and future work needs to be done on whether MRI or other imaging modalities are useful for predicting long-term functionality. Other structural neuroimaging methods include diffusion tensor imaging, susceptibility-weighted imaging, functional MRI, and perfusion imaging through arterial spin labeling [42].

TAI cannot be perceived with CT, and sensitivity is low for standard MRI. However, an MRI technique known as diffusion tensor imaging, which uses the diffusion of water molecules as they move along the nerves to visualize and trace the fiber tracts in white matter, can detect TAI [43]. The microstructural damage detected by diffusion tensor imaging and reflecting TAI is believed to be a key factor in predicting prolonged symptoms after concussion or mild TBI [44].

Figure 1 summarizes the key factors involved in TBI assessment.

### 2.4. Evaluating Concussion: Emphasis on Sports-Related Concussion

Those who participate in sports are known to be vulnerable to concussion, and this affords a unique opportunity for the collection of pre- and post-concussion data and monitoring. Sports-related concussions are a special situation because they have been the focus for the development of protocols for evaluation and practices on return to play post-concussion [45]. Sports-related concussions are frequent and tend to occur in younger populations, including adolescents and children [46]. They are considered a mild form of TBI, and the lack of consensus on the diagnostic criteria of sports-related concussion contributes to a poor pathophysiological understanding of the injury [47]. Various neurocognitive tests such as the Canadian Computed Tomography Head Rule, Immediate Post-concussion Assessment and Cognitive Testing (ImPACT), the King–Devick test, and Sports Concussion Assessment Tool (SCAT) are sensitive indicators for sports-related concussion when used shortly after the injury [48,49,50].

The Canadian CT Head Rule is a validated clinical decision algorithm designed to reduce unnecessary CT scan utilization after head injury [51,52] The five high-risk criteria on which the score is based are as follows: GCS (score lower than 15 at 2 h post-injury), skull fracture (open or depressed), signs of basal skull fracture, vomiting 2 or more times, aged 65 years and above.

Some symptoms for diagnosing acute sports-related concussion based on the 5th International Conference on Concussion in Sport include somatic symptoms such as a headache, cognitive symptoms such as feeling foggy, emotional lability, as well as physical signs such as loss of consciousness, amnesia, neurological deficit, gait unsteadiness, and drowsiness [53]. These symptoms as well as the neuropsychological assessments previously mentioned are used for clinical management, while physiological change after sports-related concussion can be analyzed in a research context using the neuroimaging techniques previously discussed. Also warranting attention is the increased risk of long-term consequences with a history of repetitive trauma to the head, which is seen in athletes who compete over the years and thus put themselves at risk of multiple concussions [54].

## 3. Point-of-Care Cognitive Tests for Mild TBI

Point-of-care cognitive testing assessment methods for mild TBI that do not involve a blood draw are readily available and often employed. The King–Devick is a well-known, inexpensive, easy-to-administer test often used on athletes [55,56]. The test takes only minutes to complete and measures rapid number reading and naming, which broadly indicates the state of cognitive function, visual–motor coordination, and language. Symptom checklists such as the Post-Concussion Symptom Scale and the SCAT as well as brief paper-and-pencil tests such as the Standardized Assessment of Concussion are also easy to administer [57,58]. The Montreal Cognitive Assessment (MoCA), which contains 12 individual tasks grouped into seven cognitive domains: (1) visuospatial/executive; (2) naming; (3) attention; (4) language; (5) abstraction; (6) memory, and (7) orientation, has also been used to screen TBI patients [59,60].

Computerized testing is also being employed, and there is flexibility in the administration of this type of evaluation as it may be done either supervised or unsupervised [61]. The CogState Brief Battery (CBB) is a computerized test that measures four core cognitive domains: processing speed, attention/vigilance, visual learning, and executive function [62,63]. All of the cognitive testing modalities described, while useful and practical at point of care, are insufficient for the diagnosis of mild TBI and can only be considered in combination with concussion history, medical history, symptoms, physical examination, and other modes of evaluation.

## 4. Blood Biomarkers for Mild TBI

The Food and Drug Administration (FDA) has given clearance to Abbott’s Alinity i TBI test, a commercially available blood test to evaluate and triage concussion patients in an acute setting. What does this really mean for clinical medicine? The FDA previously approved Abbott’s i-STAT TBI Plasma test in 2021 [64,65]. The purpose of these tests is to screen patients aged 18 years and above who present with an acute head injury and deter-mine without delay in which patients a CT scan of the brain is warranted. Since CT scanning does not detect the microstructural damage of mild TBI, it is often unnecessary, and thus the new Alinity i TBI test can be cost-efficient in reducing wasteful scanning while also benefiting patients by averting unwanted radiation exposure [66,67].

Although the radiation-sparing and expense-saving goals aspired to are commendable and consequential, the execution has suffered hindrances, mostly based on the entrenched nature of CT usage in emergency departments. Despite approval of Abbott’s i-STAT TBI Plasma test, it is unclear whether emergency department physicians have changed their use of head CTs. Resistance to the adaptation of blood biomarkers for concussion evaluation may be a major obstacle. Whether or not Abbott’s Alinity i TBI test or any blood biomarker can transform mild TBI assessment is not yet known [68,69].

The handheld point-of-care analyzer needed for Abbot’s i-STAT TBI Plasma test is expensive and may not be on hand in many hospitals [70,71]. Further, a study by Larcher et al. found that its results for some blood measures had a level of imprecision that should cause caution in interpretation [72]. In contrast, CT scanners are standard and readily available in the acute care setting, making ordering a CT scan the easiest course to follow. A negative CT gives the health professionals in the emergency department confidence that they can rule out bleeding in the brain, so the patient can then be discharged. According to Abbott, the Alinity i TBI test has a 96.7% sensitivity and complements the company’s i-STAT TBI plasma test, a rapid blood test for concussion [13,73]. Clinicians can receive a reliable result from the test in 18 min, with a 99.4% negative predictive value. A blood test for a head injury would need widespread acceptance in the medical world as a reasonable test to preclude significant brain injury.

While there are robust data in the medical literature, a huge investment in medical education might be necessary to alter standard emergency department practice patterns. Convenience is also critical. The CT scanner is already situated in the hospital, and the radiology staff remains on-site with scan interpretation accessible in person or by virtual means. There is no new investment in equipment or personnel needed. The results are intuitively reassuring; if the radiologist clears the scan, the patient can be discharged. Alternatively, blood tests for brain injury may not be as obviously and tangibly convincing for clinicians who may still feel compelled to perform a CT of the head to rule out a brain bleed.

What is the new hope presented with the recent approval of Abbott’s Alinity i TBI test? The new test can be performed on Abbott’s Alinity i laboratory instrument, as an alternative to the previous i-STAT TBI Plasma test. The Alinity i instrument is in wide use in hospitals within and beyond the USA for its ability to measure a variety of hormones, antigens, and other compounds using a chemiluminescent microparticle immunoassay [74]. This obviates the need to purchase a new instrument dedicated to the TBI test. A centrifuge is required to separate plasma from whole blood. Expenditure reduction based on estimates of the number of neuroimaging procedures avoided will be greater because the analyzer itself does not add to the cost of the TBI test.

The two biomarkers measured in the simple Alinity i TBI blood test are glial fibrillary acidic protein (GFAP), the key intermediate filament protein in astrocytes, and ubiquitin C-terminal hydrolase L1 (UCH-L1), a neuronal enzyme that marks proteins for degradation. Both of these are released by the brain into the peripheral blood when the brain is injured [75,76,77]. Multiple studies have verified the utility of these biomarkers if the head injury is evaluated within 12-24 h after injury, and there is accumulating evidence that their continued presence in plasma after TBI predicts a greater probability of neurobehavioral sequelae [64,78,79,80]. The chemical markers rapidly clear from the blood with a half-life of less than 48 h [81]. While many head injury patients are seen within this window, others may be delayed beyond 24 h, and a portion are never evaluated by a medical professional or the diagnosis is missed [82,83]. Also of special concern is the subset of patients with a history of multiple mild TBI events. All of these groups merit evaluation and longitudinal follow-up.

At this time, the only FDA-cleared in vitro diagnostic test for concussion is Abbott’s immunoassay for GFAP and UCH-L1. Other potential biomarkers may reach the market in the future. One such marker, calcium-binding protein S100, beta isoform (S100β), is a cytosolic calcium-binding protein of largely astroglial origin that enters the bloodstream after TBI. It has a half-life of under 100 min after mild TBI, and therefore, elevated serum levels over hours can be indicative of ongoing release from damaged tissue [84,85]. It is approved for use in Scandinavian countries for TBI assessment [86,87,88]. A recent retrospective study in the emergency department of a single institution showed that implementation of the S100β plasma assay in the evaluation of mild TBI safely reduced the use of imaging [89]. Another candidate TBI biomarker is neuron-specific enolase, a glycolytic enzyme found primarily in neuronal and neuroendocrine tissues that is upregulated upon neuronal cell body damage [90,91].

Amyloid protein has also caught the attention of TBI researchers because of its known relationship to neuronal damage and neurodegeneration, particularly in Alzheimer’s disease [92]. There are data to support an increase in plasma beta-amyloid after concussion and particularly after exposure to shock-wave pressure, but the use of amyloid as a TBI biomarker is in very early stages [93,94]. Amyloid-producing enzymes may also increase after TBI [95]. Murine studies also show a relationship between TBI and amyloid increase [96]. Experiments mimicking mild TBI via mechanical deformation of cultured human pluripotent stem-cell-derived neurons showed that mild axonal deformation caused amyloidogenic cleavage of axonal amyloid precursor protein with accumulation of amyloid precursor protein in the axon occurring early and continuing up to 24 h along with elevated secretion of beta-amyloid protein [97,98].

Of special note are the possible advantages of a biomarker that can distinguish TAI in mild TBI, particularly if therapies are developed for the high-risk subset of mild TBI patients with TAI to avert long-term sequelae. The neurofilament light chain, a component of the axonal cytoskeleton, is released into the cerebrospinal fluid and blood upon axonal injury. [99,100]. Since blood levels are very low and CSF measurement is invasive, the development of highly sensitive Single-Molecule Assay technology has now made neurofilament light chain measures in blood feasible, and serial measurements indicate that levels remain high over long periods after injury [101,102,103,104]. However, a recent study showed that a relationship between neurofilament light and outcome was only found in moderate-to-severe TBI and not in mild TBI [105].

Tau protein, found localized in the axon where it binds to axonal microtubules, is another candidate biomarker for axonal injury in TBI [106,107]. Plasma levels of phosphorylated forms of tau are increasingly being used in the diagnosis of Alzheimer’s disease, and levels of phosphorylated tau and total tau rise acutely after mild TBI [108,109,110,111]. Eagle et al. found that in mild TBI patients with symptoms lingering past three months post-trauma, higher plasma levels of phosphorylated tau pre-intervention predicted poorer response to targeted intervention for their chronic symptoms [112]. UCH-L1 was also a significant predictor of improvement in this small study of 74 participants.

## 5. Potential Expanded Use of Point-of-Care Biomarker TBI Testing

Persons who have suffered a TBI and reach an emergency department or other facility equipped with the analyzer within the delimited time period are considered candidates for Abbott’s new Alinity i TBI test. The biomarker levels in these patients could serve as a source of valuable data, whether or not the CT scan is obtained in tandem. The presence or levels of GFAP and UCH-L1 cannot reliably discriminate between patients who will improve rapidly versus those with prolonged recovery time who may continue to experience troubling neurologic symptoms [113]. Currently, there is no test with that capability [114,115]. This poses a challenge, especially as a majority of TBI patients have no follow-up scheduled. A positive blood test might suggest that follow-up care is indicated, but we propose that TBI biomarker data obtained where healthcare is provided such as the Alinity i TBI test can give the medical community so much more. Analysis and integration of data aggregated in emergency departments performing TBI biomarker testing could produce a massive database that would allow the development of a predictive model so that those at the highest risk for poor recovery could be targeted for early intervention. Sports provide a unique advantage in data collection with an opportunity to obtain preseason baseline tests so that within-player change in plasma biomarkers with TBI could be accurately captured [116].

In addition, the lack of effective treatment for mild TBI beyond rehabilitation and symptomatic relief makes the idea of applying the Alinity i TBI test data to acquire a more complete understanding of damage development very appealing [117,118]. This type of understanding is necessary if a breakthrough is to come, whether through the repurposing of current drugs or the application of new gene-expression-altering technologies to interrupt a program of neuronal destruction set in motion by the head injury [119]. While some false positives and false negatives are inevitable, validation procedures and other statistical methodologies could optimize the accuracy of the results from such a large number of data points sourced from many patients [120,121]. The lack of progress or innovation in current TBI research leaves an obvious and urgent need for a way forward that might be filled using new blood-based biomarkers.

Yet another consideration in biomarker use in TBI is the lack of any validated assessment for those under age 18. The absence of representation of the pediatric concussion patient is a major gap since the developing brain is particularly susceptible to injury and prolonged sequelae [122,123]. In older persons, baseline levels of biomarkers are higher, and discriminative capability may be less or of shorter duration after injury [115,124,125]. In persons with Alzheimer’s disease, GFAP levels may be elevated [126]. The need to devise age-appropriate TBI biomarker tests is clear.

The value of the Alinity I test in TBI patients who present after the recommended testing window is somewhat in question. Although biomarker levels lose their ability to distinguish who should receive imaging, that does not preclude using the data to develop a model that will give us prognostic information concerning the chance that long-term neurologic consequences will occur [64,127,128]. Further, differences in the trajectory of biomarker changes might be different after a second or third injury in those with prior TBI history, and this may provide information on resilience and factors that affect recovery after repeated injury [129]. While these types of subtle differences might not be detected when studying hundreds of patients, informative patterns and relationships may surface when information from tens of thousands of concussion patients is profiled and combined [64]. Although severe TBIs are usually recognized and treated, a majority of mild injuries are unrecognized [130,131]. While there is currently no effective pharmacologic intervention in the vast majority of mild TBI cases and this continues to be a major gap in medical knowledge, the confirmation of a concussion leads to a recommendation for a period of rest followed by a gradual return to full activity [132,133]. This guidance is designed to prevent long-term post-concussion symptoms and could be applied more accurately if the Alinity i TBI test data as well as any other TBI biomarker assay data from new systems gaining FDA approval were used to refine our ability to diagnose mild TBI.

Abbott’s Alinity i TBI blood test may be beneficial when screening brain injury patients for head CTs, a decision that will be made by the treating healthcare professional. However, untapped benefits of a large biomarker and outcome resource providing adequately powered samples for validation and prediction could arise from this and other future commercially available mild TBI blood tests. This could bring a new chapter in brain injury evaluation and treatment. This could prompt refinement in rehabilitation protocols as well as the development of new pharmacotherapies where the need is great and the options are lacking [134]. Studies of this type are in progress now on a smaller scale, but the handheld systems make expansion practical [99,135]. Large databases and electronic medical records make it easy to operationalize this new type of investigation in order to offer better, more targeted care and a brighter future for those experiencing the aftereffects of mild TBI [136].

## 6. Conclusions

There is not a clear consensus on diagnostic assessments for TBI, and CT scans are often used because of their availability in acute settings. As described, CT scans are useful for visualizing the presence and location of hemorrhages but are not as sensitive as MRIs for axonal injury visualization. It is also unclear how clinically useful these neuroimaging techniques are for long-term prognosis and quality of life. Abbott’s Alinity i TBI test may be beneficial in evaluating common biomarkers released upon brain injury such as GFAP and UCH-L1. The newfound increasing availability of Abbott’s Alinity i laboratory instrument contrary to the previous i-STAT TBI Plasma test demonstrates the potential cost-effectiveness of the test and its usefulness in detecting lasting brain damage. This development brings a new opportunity for TBI research to grow and make useful conclusions for long-term physiological and neurocognitive prognosis. The ultimate goal would be to use aggregate data to pinpoint those mild TBI patients with the highest risk of long-term debilitating neurologic problems and then implement mitigating actions to avert these consequences.

## Figures and Tables

**Figure 1 diagnostics-13-03330-f001:**
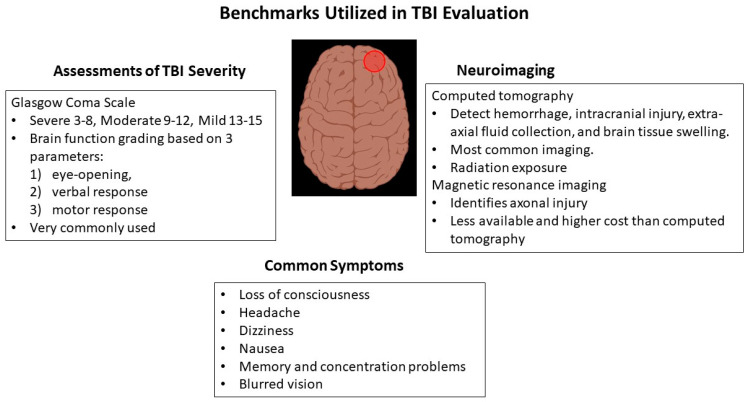
Major components in the assessment of TBI severity. Red area represents site of injury.

## Data Availability

Not applicable.

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
