# Peer review of "Maximizing the Clinical Value of Blood-Based Biomarkers for Mild Traumatic Brain Injury"

_diagnostics, 2023, doi:10.3390/diagnostics13213330_

Round 1
Reviewer 1 Report
Comments and Suggestions for Authors
The manuscript provides an overview of the diagnosis of mild traumatic brain injury especially about the blood-based biomarkers. There are some suggestions as follows:
1. Introduction (Line 39): The introduction of this manuscript provides a comprehensive overview of the topic. However, it would be beneficial to clarify the connection between mild traumatic brain injury (mTBI) and traumatic axonal injury (TAI) when mentioning TAI in this context. While TAI is an important aspect of TBI, it might be helpful to explain how it relates specifically to mTBI in the context of blood-based biomarkers.
2. Section 2.4 - Sports-Related Concussions: This section appears to be somewhat disjointed from the rest of the manuscript. While the discussion of sports-related concussions is relevant, it could be better integrated into the broader context of diagnosis methods for mTBI. Furthermore, the discussion of symptoms in lines 133-142 would be more appropriately placed within a section dedicated to the clinical presentation or symptoms of mTBI rather than being isolated in this manner.
3. Abbott i-STAT TBI Plasma Test: While the authors have provided valuable information regarding the Abbott i-STAT TBI plasma test, it seems somewhat disproportionate to focus extensively on this one point-of-care testing method without mentioning other relevant alternatives. A more balanced approach that acknowledges a broader spectrum of point-of-care testing methods for mTBI diagnosis would enhance the comprehensiveness of this review.
Author Response
We thank the reviewer for thoroughly scrutinizing our manuscript. As requested, we have revised the manuscript and addressed the specific comments of the reviewer. The revised sections are delineated in red in a marked copy of the manuscript text.
Below, we provide a point-by-point response to the reviewer’s comments.
Reviewer # 1 Comments
- COMMENT #1: Introduction (Line 39): The introduction of this manuscript provides a comprehensive overview of the topic. However, it would be beneficial to clarify the connection between mild traumatic brain injury (mTBI) and traumatic axonal injury (TAI) when mentioning TAI in this context. While TAI is an important aspect of TBI, it might be helpful to explain how it relates specifically to mTBI in the context of blood-based biomarkers.
RESPONSE: We have added discussion of TAI in 3 places: in introduction, in imaging and in biomarker sections.
- COMMENT #2: Section 2.4 - Sports-Related Concussions: This section appears to be somewhat disjointed from the rest of the manuscript. While the discussion of sports-related concussions is relevant, it could be better integrated into the broader context of diagnosis methods for mTBI. Furthermore, the discussion of symptoms in lines 133-142 would be more appropriately placed within a section dedicated to the clinical presentation or symptoms of mTBI rather than being isolated in this manner.
RESPONSE: This subsection, within the section “Current Modes of Mild TBI Evaluation” has been renamed “Evaluating Concussion: Emphasis on Sports-related Concussion” and further explanation of the reasons for addressing sports-related concussion specifically have been added.
- COMMENT #3: Abbott i-STAT TBI Plasma Test: While the authors have provided valuable information regarding the Abbott i-STAT TBI plasma test, it seems somewhat disproportionate to focus extensively on this one point-of-care testing method without mentioning other relevant alternatives. A more balanced approach that acknowledges a broader spectrum of point-of-care testing methods for mTBI diagnosis would enhance the comprehensiveness of this review.
RESPONSE: We agree and have added a new section entitled “Point-of-care Cognitive Tests for Mild TBI” (Now section 3).

Reviewer 2 Report
Comments and Suggestions for Authors
The authors present these biomarkers for Mild Traumatic Brain Injury, but There are other less well-known biomarkers in literature.
These Biomarkers should still be mentioned, given the importance of having blood biomarkers in emergency.
Author Response
We thank the reviewer for thoroughly scrutinizing our manuscript. As requested, we have revised the manuscript and addressed the specific comments of the reviewer. The revised sections are delineated in red in a marked copy of the manuscript text.
Below, we provide a point-by-point response to the reviewer’s comments.
Reviewer # 2 Comments
- COMMENT #1: There are other less well-known biomarkers in literature. These Biomarkers should still be mentioned, given the importance of having blood biomarkers in emergency.
RESPONSE: We agree and have added to the discussion. We have expanded discussion of S100β and now include neurofilament light chain and tau.

Round 2
Reviewer 1 Report
Comments and Suggestions for Authors
The authors revised their manuscript according to the previous comments.
Reviewer 2 Report
Comments and Suggestions for Authors
Accept in this form